# Supercritical Fluid Technology for the Development of 3D Printed Controlled Drug Release Dosage Forms

**DOI:** 10.3390/pharmaceutics13040543

**Published:** 2021-04-13

**Authors:** Johannes Schmid, Martin A. Wahl, Rolf Daniels

**Affiliations:** Department of Pharmaceutical Technology, Eberhard Karls University, Auf der Morgenstelle 8, 72076 Tuebingen, Germany; mail@johannes-schmid.eu (J.S.); martin.wahl@uni-tuebingen.de (M.A.W.)

**Keywords:** 3D printing, supercritical, carbon dioxide, controlled particle deposition, controlled drug release

## Abstract

Supercritical CO_2_ loading of preformed 3D printed drug carriers with active pharmaceutical ingredients (APIs) shows great potential in the development of oral dosage forms for future personalized medicine. We designed 3D printed scaffold like drug carriers with varying pore sizes made from polylactic acid (PLA) using a fused deposition modelling (FDM) 3D printer. The 3D printed drug carriers were then loaded with Ibuprofen as a model drug, employing the controlled particle deposition (CPD) process from supercritical CO_2_. Carriers with varying pore sizes (0.027–0.125 mm) were constructed and loaded with Ibuprofen to yield drug-loaded carriers with a total amount of 0.83–2.67 mg API (0.32–1.41% *w*/*w*). Dissolution studies of the carriers show a significantly decreasing dissolution rate with decreasing pore sizes with a mean dissolution time (MDT) of 8.7 min for the largest pore size and 128.2 min for the smallest pore size. The API dissolution mechanism from the carriers was determined to be Fickian diffusion from the non-soluble, non-swelling carriers. Using 3D printing in combination with the CPD process, we were able to develop dosage forms with individually tailored controlled drug release. The dissolution rate of our dosage forms can be easily adjusted to the individual needs by modifying the pore sizes of the 3D printed carriers.

## 1. Introduction

Supercritical fluid technology offers a wide range of opportunities in drug development and production due to its unique properties. CO_2_ is most commonly used for supercritical applications [1], since it is non-toxic and reaches supercritical conditions at a relatively low pressure of 7.4 MPa near room temperature (304 K) [2]. With a density near liquids and a viscosity near gases, supercritical CO_2_ shows excellent diffusivity and mass transfer capabilities [3,4]. These properties can be utilized for extractions of plant materials, where the extractive power can easily be adjusted by controlling the pressure and temperature of the extraction system, which gives this technique an extra degree of freedom, compared to liquid extractions [3,5]. In the pharmaceutical sector, supercritical fluid technology can be applied to several different applications, like particle and crystal engineering, composite particles preparation, coating of solid dosage forms, liposome preparation, or protein extraction and drying [6]. For the micronization of poorly soluble active pharmaceutical ingredients, several processes based on supercritical fluids were developed [7], such as rapid expansion of supercritical solutions (RESS), where a supercritical fluid gets saturated with a substrate of interest following a sudden depressurization through a nozzle, which results in rapid precipitation of the solute [8]. Based on this process, the controlled particle deposition process (CPD) was developed, where a drug is dissolved in supercritical CO_2_ followed by penetration of the supercritical solution into the pores of the carrier and precipitation of the drug inside the pores by rapid pressure reduction. With this technique, Türk et al. were able to achieve an almost complete inclusion of ibuprofen in β-cyclodextrin, which resulted in a significantly higher dissolution rate than that of untreated ibuprofen and its physical mixture with β-cyclodextrin [9]. Wischumerski et al. were able to apply the CPD method to preformed porous oral dosage forms, which also showed a significantly faster dissolution rate compared to untreated ibuprofen powder [10].

3D printing has found its way into pharmaceutics and medicine recently and has seen a significant boost over the past few years. This technique allows one to obtain a solid object from a 3D model, realized with 3D modelling software. The 3D printed product is obtained using an additive process, in which layers of material are laid down one over the other successively [11]. Many different 3D printing technologies like binder jet printing, fused deposition modelling (FDM), selective laser sintering (SLS), stereolithography (SLA) and pressure assisted micro syringe (PAM) are subject to current pharmaceutical and medical research [12,13,14,15,16]. With these technologies, numerous 3D printed products for pharmaceutical or medical applications have been realized up to today, like surgical guides and simulation models; customized implants and endoprostheses; bone, skin and cartilage tissues for implantation; scaffold-like structures for tissue regeneration; subcutaneous drug eluting implants; and a variety of oral dosage forms [12,13,14,15,17,18,19,20,21]. The FDM printing method has proven to be a suitable technique for the development and production of 3D printed oral dosage forms. Immediate release tablets [22,23], gastro retentive tablets [24], bilayer tablets [25], tablets with modified release characteristics [26,27,28] and many more manufactured in the FDM 3D printing process are described in the literature. These dosage forms require the manufacturing of API-loaded filaments prior to printing, which is being achieved by hot melt extrusion or filament impregnation with organic solvents. Furthermore, the necessity of high printing temperatures of the drug-loaded filaments presents a major challenge for the production of 3D printed dosage forms with temperature-sensitive drugs, like atorvastatin [29] or amlodipine [30].

Combining the technologies of 3D printing and drug loading via controlled particle deposition offers many advantages since it uses a solvent-free low-temperature process for the production of the drug-loaded oral dosage forms. The aim of this study was to assess 3D printed oral dosage forms that were API loaded using the CPD process subsequently. Employing a commercial FDM 3D printer, scaffold-like structures made from water insoluble polymers with varying pore sizes for the control of drug release were developed. In a second step, loading conditions for ibuprofen as a model drug were defined to yield API-loaded drug carriers. Ibuprofen as a model drug was chosen because of its good solubility in supercritical CO_2_ at our working conditions [9], which makes it suitable for scCO_2_ drug-loading applications. In respect of being beneficial for application in personalized medicine, this process was evaluated regarding the influence of the 3D carrier models on the drug-loaded carrier properties. Thus, four differently designed 3D printed drug carriers were developed to determine the API dissolution rate depending on the carrier’s pore sizes.

## 2. Materials and Methods

### 2.1. Chemicals

PLA filament (transparent premium PLA) was obtained from German RepRap GmbH, Feldkirchen, Germany. Ibuprofen was purchased from Vivatis Pharma GmbH, Hamburg, Germany. CO_2_ (Technical grade ≥ 99.5%) was obtained from Westfalen AG, Muenster, Germany. Potassium dihydrogen phosphate was obtained from Carl Roth GmbH and Co. KG, Karlsruhe, Germany. Sodium hydroxide was purchased from chemical supplies pharmaceutical institute, University of Tuebingen, Tuebingen, Germany. Tween 80 was obtained from Croda International, GB-East Yorkshire, Great Britain. Methanol (HPLC gradient grade) was purchased from Merck KGaA, Darmstadt, Germany. Highly purified water was produced using a Purelab Option Q7 (Veolia Water Technologies Deutschland GmbH, Celle, Germany).

### 2.2. Development of 3D Printed Drug Carriers

The first step of 3D printing of porous drug carriers was the development of 3D computer models using the AutoCAD 2019 3D modelling software (Autodesk GmbH, Munich, Germany). Figure 1 shows the schematic structure of the models. The carriers were constructed as cylindrical monoliths with a height of 6 mm and a width of 8 mm, consisting of layers with a thickness of 0.1 mm, twisted to each other by 90 degrees. Every layer was composed of paths with defined width, which were placed at a defined distance to each other. This arrangement results in horizontally and vertically occurring pores of defined height and width running through the carrier. Variations in the carriers were obtained by modifying the layer thickness, the width of the paths inside the layer and the distance between these paths. In this study, the gaps between the paths were altered in order to obtain carriers with varying pore sizes, shown in Table 1.

### 2.3. 3D Printing of the Drug Carriers

A slicer program (Simplify3D 4.0, Simplify3D, Cincinnati, OH, USA) was used to set the printing options. All carriers were printed with the same printing setting. Critical parameters of the printing setting are listed in Table 2. Drug carriers were printed in a German RepRap X350pro fused deposition modelling 3D printer with transparent premium PLA, using a 0.25 mm brass nozzle (German RepRap GmbH, Feldkirchen, Germany).

### 2.4. SEM Imaging of 3D Printed Drug Carriers

SEM imaging of 3D printed drug carriers was performed by means of a DSM 940 A scanning electron microscope (Carl Zeiss AG, Oberkochen, Germany) at an accelerating voltage of 5 kV. Before imaging, the drug carriers were sputter coated with gold using an E5100 sputter coater (Bio-Rad Laboratories GmbH, Feldkirchen, Germany). Sputter conditions can be found in Table 3.

### 2.5. API Loading of the Drug Carriers

API loading was performed in a scCO_2_ pilot plant unit (Sietec-Sieber, Maur, Switzerland) with Ibuprofen as model drug, employing the controlled particle deposition method (CPD) [1]. Figure 2 shows the simplified layout of the scCO_2_ pilot plant unit.

Figure 3 shows the scheme of the CPD process. In this process, Ibuprofen and the 3D printed carriers were placed in a pressure chamber (B1), which was filled with CO_2_ by closing V5 and opening V3. Ibuprofen concentration in the chamber was set to 12.5 mg/cm^3^ to ensure rapid precipitation when decreasing the pressure [31]. The temperature of B1 was set to 40 °C by means of a double jacket. By opening V6 for 10 s, waiting for 60 s and repeating these steps two more times, remaining air in B1 was removed. CO_2_ was then pumped into B1 until it reached a pressure of 25 MPa at 40 °C. V3 was closed to keep the pressure constant. After 20 h, the pressure was decreased rapidly, allowing for the API to deposit inside the porous carrier. Because of the Joule–Thompson effect, the temperature in B1 strongly decreases during the rapid pressure release. Therefore, the drug-loaded carriers need to remain in B1 for 45 min after the pressure decrease, to prevent condensation of water on the carriers.

### 2.6. Confocal Raman Microscopic Analysis of Loaded Drug Carriers

Spectra of the drug carriers were acquired using an alpha 500R confocal Raman microscope (WiTec GmbH, Ulm, Germany) equipped with a 532 nm excitation laser, UHTS 300 spectrometer, a 600 gr/mm grating and DV401-BV CCD camera. A 40x/0.6 NA objective was used (EC Epiplan-neofluor; Carl Zeiss AG, Oberkochen, Germany) in combination with a 50 μm optical fibre. Spectra were recorded on the surface of the loaded drug carriers in an area of 150 × 150μm with a step size of 3 µm. Integration time was 0.5 s. Laser power was 25 mW. Spectra of ibuprofen and an unloaded drug carrier were used to calculate reference spectra. These spectra were used to determine the spatial distribution of ibuprofen on the surface of drug-loaded carriers. All spectra were processed by cosmic ray removal and baseline correction with the software Project Plus 4 (WiTec GmbH, Ulm, Germany)

### 2.7. Dissolution Studies

Dissolution was performed in an DT-D6 dissolution tester (ERWEKA GmbH, Langen, Germany) with a Lauda immersion thermostat T (Lauda Dr. R. Wobser GmbH and Co. KG, Lauda-Koenigshofen, Germany) at sink conditions according to the requirements of the European Pharmacopoeia 10.2 using the basket method (2.9.3. Dissolution test for solid dosage forms, Apparatus 1). Phosphate buffer pH 7.2 with an addition of 0.1% Tween 80 was used as dissolution medium (Buffer recipe is shown in Table 4). The rotation speed was set to 50 rpm, and the temperature was set to 37 °C. Samples were analyzed via HPLC (see chapter 4.5). API loading capacity of the drug-loaded carriers was determined via the overall released API amount in the dissolution studies. For a better visual comparison of the dissolution profiles, the released API amount is presented as percentage of the overall released API amount.

### 2.8. Evaluation of Dissolution Test

To compare the dissolution rates of the different loaded carriers, the mean dissolution time MDT for each carrier was calculated in Equation (1)
(1)$$\mathrm{MDT}=\frac{\sum_{i=i}^{n}{\bar{t}}_{i}\cdot \Delta M_{i}}{\sum_{i=1}^{n}\Delta M_{i}}$$
with ${\bar{t}}_{i}$ being the midpoint of the sampling interval and $\Delta M_{i}$ being the API amount dissolved in that interval [32]. MDT values were statistically analyzed using a one-way ANOVA with Tukey post-hoc test.

The Korsmeyer–Peppas Equation (2) was used to estimate the dissolution mechanism.
(2)$$\frac{M_{t}}{M_{\infty }}=kt^{n},\;\mathrm{for}\;\frac{M_{t}}{M_{\infty }}\;<\;60\%$$

In this equation, $M_{t}$ is defined as the amount of API released at time t, $M_{\infty }$ is the overall dissolved API amount and n is the diffusional exponent. Transforming Equation (2) yields the linearized Equation (3).
(3)$$\log\left(\frac{M_{t}}{M_{\infty }}\right)=n\cdot \log\left(t\right)+\log\left(k\right)$$

The logarithm of the relative dissolved API amount plotted against the logarithm of time yields a straight line with the slope n. The diffusional coefficient n (n-exponent value) describes the drug release mechanism as shown in Table 5 [33].

### 2.9. Ibuprofen Assay

HPLC analysis was performed using a Shimadzu LC-20AT HPLC system (Shimadzu Europa GmbH, Duisburg, Germany) with an EC 125/4 Nucleosil 100-5 C18 column in combination with a Nucleosil 100-5 C8CC 8/3 precolumn (Macherey-Nagel GmbH and Co. KG, Dueren, Germany). As mobile phase, methanol:phosphate buffer 20 mM, pH 3 (70:30) was used at a constant flow rate of 1 mL min^−1^. Column oven temperature was set to 25 °C. For each sample 20 µL was injected, and the UV absorbance was measured at 264 nm. Ibuprofen was eluted after ~4.2 min. The peak area was used to calculate the ibuprofen concentration of the samples using a calibration curve in the range of 1.6 to 16 µg mL^−1^. The limit of detection and the limit of quantification were determined to be 0.299 µg mL^−1^ and 0.905 µg mL^−1^, respectively.

## 3. Results

### 3.1. Visual Comparison of 3D Printed Drug Carriers

After the 3D printing process, a visual comparison of the different drug carrier types was performed (Figure 4). All carriers proved to be visually similar, having the same height and diameter and differing only in the distance between the material paths resulting in different pore sizes. Moreover, all carriers were printed with the same printing settings shown in Table 2. Using a scanning electron microscope (SEM), the carrier scaffold structure can be displayed in more detail. Figure 5, Figure 6, Figure 7 and Figure 8 show SEM pictures of the different carrier models. Starting with carrier 1, which was designed to have the largest pores, the pictures show decreasing pore sizes with rising carrier numbers.

### 3.2. Confocal Raman Microscopic Analysis of Loaded Drug Carriers

Figure 9 shows a false color image of a drug-loaded carrier’s surface in direct vicinity to a pore in the carrier. Ibuprofen presence on the carriers’ surface was determined via the characteristic ibuprofen peak at a wavelength of ~1610 cm^−1^. This proves the presence of Ibuprofen on the surface of drug-loaded carriers. Additionally, the false color image shows a homogeneous distribution of ibuprofen across the surface of the 3D printed drug carrier.

### 3.3. Dissolution Studies

Figure 10 shows the dissolution profiles of the carriers with decreasing pore sizes from carrier 1 to 4. The drug loading on the drug carriers was determined to be 2.67 ± 0.20 mg (carrier 1), 1.45 ± 0.09 mg (carrier 2), 1.30 ± 0.11 mg (carrier 3) and 0.83 ± 0.06 mg (carrier 4), which equals an API loading capacity on the drug carriers of 1.41 ± 0.10% (carrier 1), 0.70 ± 0.04% (carrier 2), 0.57 ± 0.05% (carrier 3) and 0.32 ± 0.02% (carrier 4). Mean dissolution time (MDT) values for the carriers were calculated to be 8.7 ± 3.3 min (carrier 1), 31.3 ± 2.9 min (carrier 2), 37.2 ± 4.7 min (carrier 3) and 128.2 ± 2.2 min (carrier 4). The difference in dissolution rate between carriers 2 and 3 was statistically not significant. Carrier 1 with the largest pore size shows a significantly faster dissolution rate than carrier 2, 3 and 4 (*p* < 0.001). Additionally, carrier 4 with the smallest pore size shows a significantly slower dissolution rate than all other carriers (*p* < 0.0001). Figure 11 visualizes the dissolution rate, expressed as MDT, depending on the pore size of the 3D printed drug carriers. The smaller the pore size of the carrier, the slower the dissolution rate.

Figure 12 shows the Korsmeyer–Peppas Plots for carriers 2, 3 and 4 with excellent correlations of R^2^ = 0.9965 for carrier 2, R^2^ = 0.9941 for carrier 3 and R^2^ = 0.9931 for carrier 4. N-exponent values were determined to be *n* = 0.4303 for carrier 2, *n* = 0.4344 for Carrier 3 and *n* = 0.4368 for carrier 4, representing Fickian diffusion from the non-soluble, non-swelling carriers as dissolution mechanism [33], explaining the extended release profiles of the API-loaded carriers. The n-exponent value could not be determined for Carrier 1 due to the fast dissolution, with more than 60% being dissolved after only 2 min.

## 4. Discussion

Combining both the technologies of 3D printing and drug loading via controlled particle deposition was shown to be a suitable process for the manufacturing of oral dosage forms with controlled drug release. Confocal Raman microscopic analysis of the loaded drug carriers showed a homogeneous distribution of Ibuprofen on the surface of the carriers without the presence of drug agglomerates or visible crystals. This was to be expected, as the CPD process was originally developed for the micronization of poorly soluble drugs with a simultaneous deposition of the drug on porous drug carriers [9,10]. By modifying the pore sizes of the carrier 3D models, we were able to achieve controlled drug release from the carriers. Different kinds of mechanisms for the control of drug release of 3D printed dosage forms are described in the literature. 3D printed tablets from co-extruded filaments consisting of an API, water insoluble polymer and an hydrophilic pore forming material show a sustained drug release that can be adjusted by modifying the amount of hydrophilic pore-forming material [27]. By API loading of water-soluble polymers via hot melt-extrusion or solvent incorporation with subsequent 3D printing of dosage forms, a sustained release due to gradual dissolution of the water-soluble polymer can be achieved [28,34]. Other approaches are using complex multilayer or multicompartment dosage forms to achieve a control of drug release [25,26,35].

By combining 3D printing of insoluble scaffold like porous carriers and drug loading with the CPD process, we were not only able to achieve a sustained release oral dosage form but also to modify the dissolution rate by varying the pore sizes of the carriers. With this, we were able to control the drug release by simply changing the 3D model, without the need to change excipients or other parameters of the manufacturing process. This opens the possibility of a fast and highly flexible manufacturing of dosage forms, tailored specifically to patients’ needs. Furthermore, in contrast to conventional 3D printed dosage forms, this process allows for a solvent-free and low-temperature processing, which allows for environmentally friendly manufacturing even for temperature-sensitive APIs. A major challenge of this process proved to be the limited drug loading on the carriers. Cerda et al. [36] achieved a drug loading of 3D printed dosage forms of close to 3% *w*/*w* using passive diffusion. Using hot melt extrusion for the manufacturing of API-loaded filaments, Ayyoubi et al. [37] were able to achieve a drug loading of up to 60% *w*/*w*. In this study, employing the CPD process to load 3D printed porous drug carriers, a drug loading of about 1.4% *w*/*w* was achieved. This represented substantially lower drug loading than conventional API loading techniques for 3D printed dosage forms, which resulted in a total API amount of less than 2.7 mg on the carriers. Nevertheless, a wide range of highly potent low-dosed APIs, like tamsulosin, could be processed using controlled particle deposition to manufacture 3D printed oral dosage forms with controlled drug release. This suggests potential future directions of this process, where the next development steps need to include highly potent APIs for the production of 3D printed oral dosage forms to achieve therapeutic API levels on the drug carriers, which will bring this technology one step closer to patients.

## 5. Conclusions

Particle deposition from a supercritical solution, employing scCO_2_ as fluid, proved to be a suitable technique to load drugs into small pore preformed carriers. As carriers, we designed cylindrical monoliths with an external size comparable to commercially available tablets (6 × 8 mm) from PLA. With decreasing pore sizes of the carriers, we were able to obtain decreasing dissolution rates. Thus, we were able to create individually tailored dissolution profiles by 3D modeling of the carriers. The dissolution mechanism of the API from the drug carriers was determined to be Fickian diffusion from the non-soluble, non-swelling carriers. The study opens the use of supercritical fluid technology in combination with 3D printing as a reliable method to prepare individually tailored oral dosage forms for controlled drug release.

## Figures and Tables

**Figure 1 pharmaceutics-13-00543-f001:**
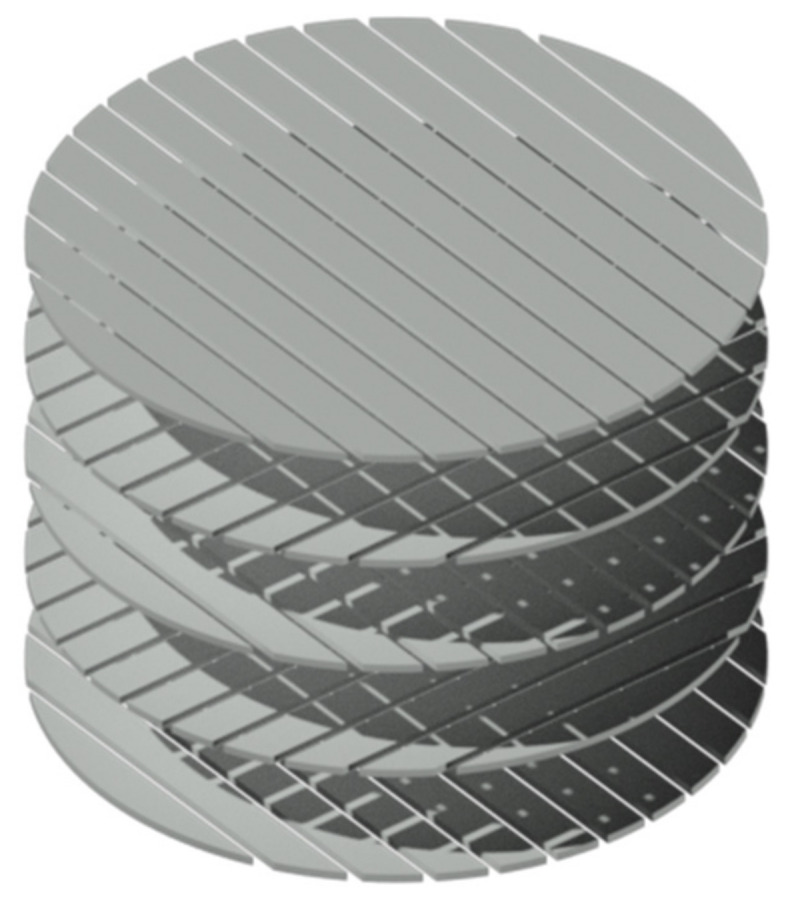
Schematic structure of a drug carrier 3D model.

**Figure 2 pharmaceutics-13-00543-f002:**
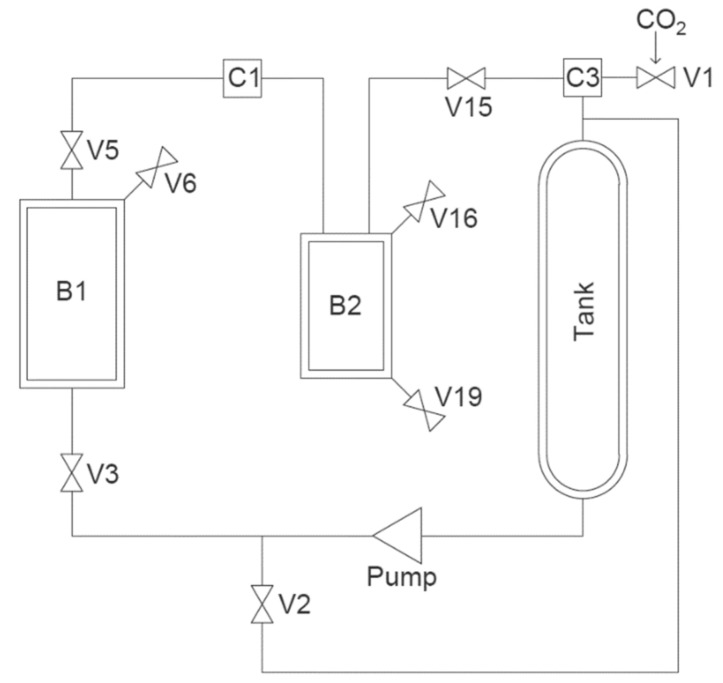
Simplified layout of the scCO2 pilot plant unit.

**Figure 3 pharmaceutics-13-00543-f003:**
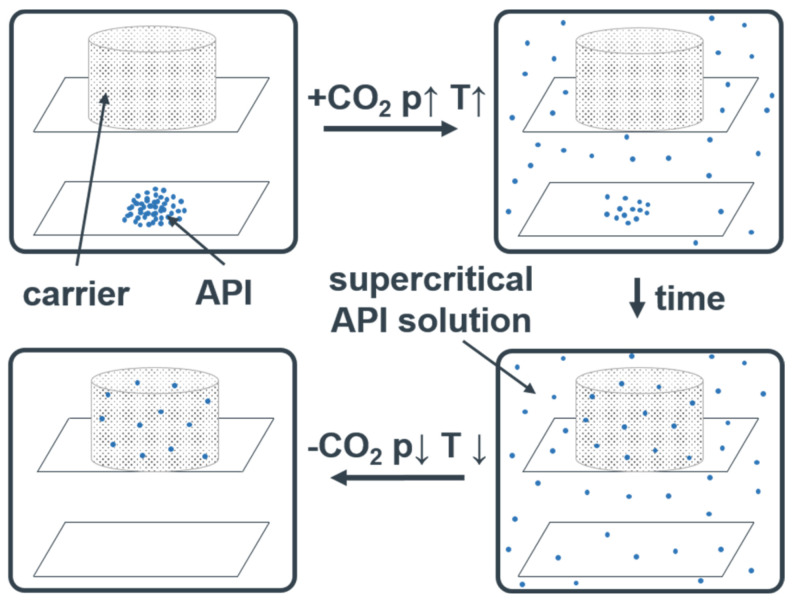
Controlled particle deposition (CPD) process.

**Figure 4 pharmaceutics-13-00543-f004:**
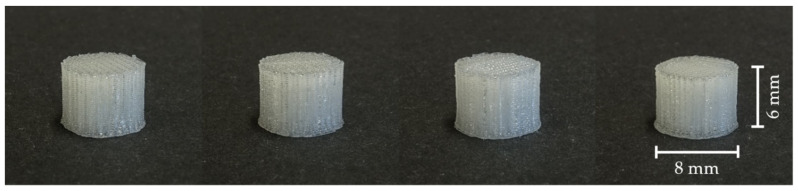
Visual comparison of 3D printed drug carriers (left to right: carrier 1 to 4).

**Figure 5 pharmaceutics-13-00543-f005:**
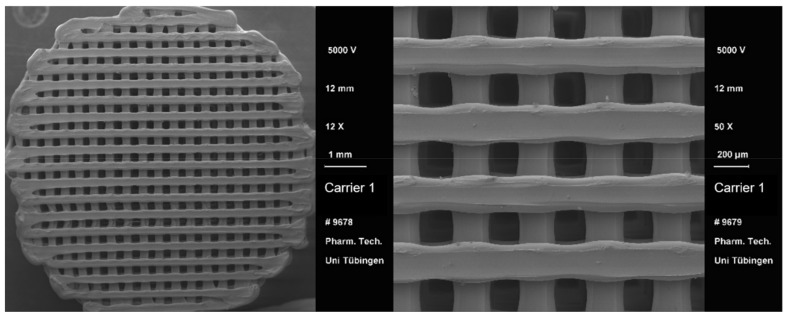
SEM pictures of carrier 1.

**Figure 6 pharmaceutics-13-00543-f006:**
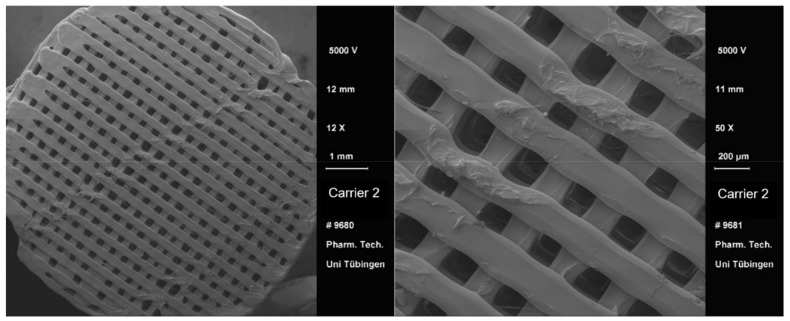
SEM pictures of carrier 2.

**Figure 7 pharmaceutics-13-00543-f007:**
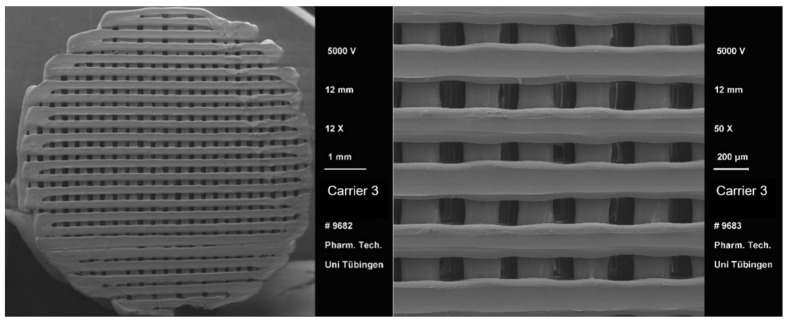
SEM pictures of carrier 3.

**Figure 8 pharmaceutics-13-00543-f008:**
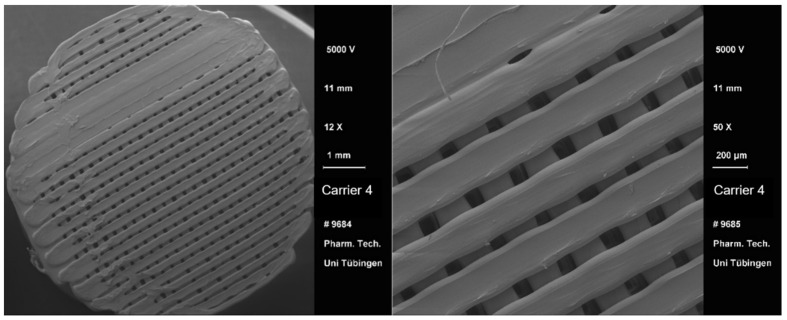
SEM pictures of carrier 4.

**Figure 9 pharmaceutics-13-00543-f009:**
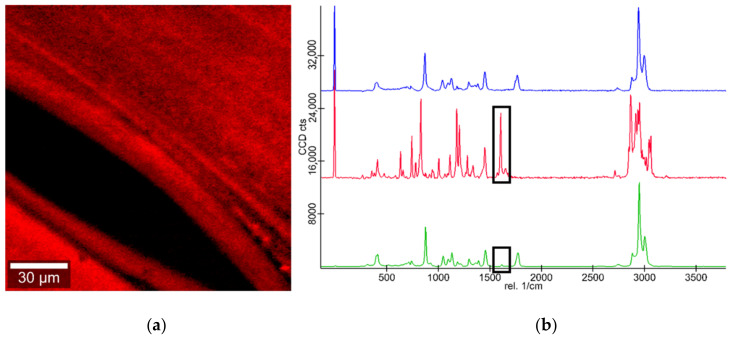
(**a**) False color image of ibuprofen on the drug carrier surface; ibuprofen on carrier surface (red); and pore (black). (**b**) Corresponding Raman spectra of PLA (blue), ibuprofen (red) and ibuprofen loaded carrier (green).

**Figure 10 pharmaceutics-13-00543-f010:**
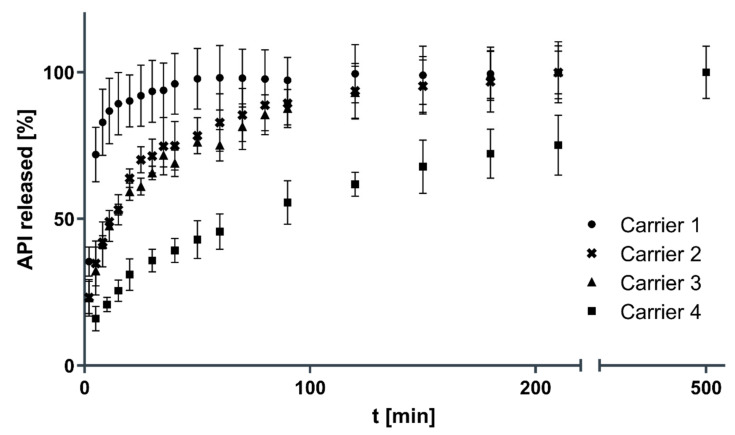
Comparison of dissolution profiles (*n* = 3, mean ± SD).

**Figure 11 pharmaceutics-13-00543-f011:**
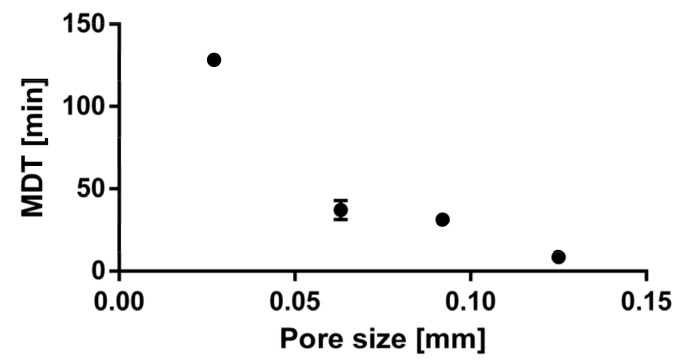
MDT dependency on the pore size (*n* = 3, mean ± SD).

**Figure 12 pharmaceutics-13-00543-f012:**
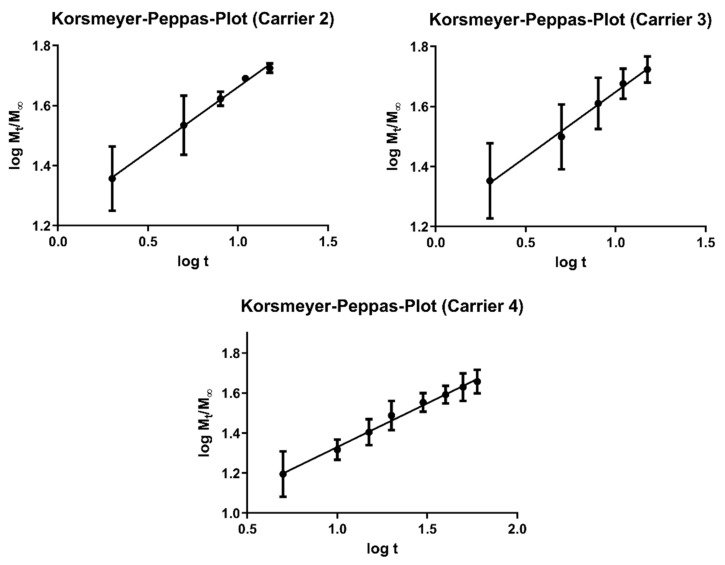
Korsmeyer–Peppas Plots for carriers 2, 3 and 4.

**Table 1 pharmaceutics-13-00543-t001:** 3D model parameters.

Model No.	1	2	3	4
Layer thickness (mm)	0.1	0.1	0.1	0.1
Path width (mm)	0.26	0.26	0.26	0.26
Distance between paths (mm)	0.125	0.092	0.063	0.027

**Table 2 pharmaceutics-13-00543-t002:** 3D printing parameters.

Printing Parameter	Setting
Printing temperature (°C)	200
Printing bed temperature (°C)	58
Nozzle diameter (mm)	0.25
Layer thickness (mm)	0.10
Extrusion width (mm)	0.26
Extrusion multiplier	0.80
Retraction distance (mm)	1.00
Retraction speed (mm s^−1^)	100
Printing speed (mm s^−1^)	13.0
x/y movement speed (mm s^−1^)	13.0

**Table 3 pharmaceutics-13-00543-t003:** Sputter coating settings.

Sputter Parameter	Setting
Vacuum (mbar)	0.04
Current (mA)	20
Accelerating voltage (kV)	2.1
Sputter time (s)	4 × 60

**Table 4 pharmaceutics-13-00543-t004:** Phosphate buffer pH 7.2 recipe.

Substance	Amount
0.2 M Potassium dihydrogen phosphate R (mL)	250.0
0.2 M Sodium hydroxide (mL)	175.0
Tween 80 (g)	1.0

**Table 5 pharmaceutics-13-00543-t005:** Diffusional exponent and diffusional release from non-swellable release systems.

Thin Film	Cylindrical Sample	Spherical Sample	Drug Release Mechanism
n = 0.50	n = 0.45	n = 0.43	Fickian diffusion
0.50 < n < 1.00	0.45 < n < 1.00	0.43 < n < 1.00	Anomalous (non-Fickian) transport
n = 1.0	n = 1.0	n = 1.0	Zero order release

## Data Availability

Data sharing is not applicable to this article.

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
