# Peer review of "Supercritical Fluid Technology for the Development of 3D Printed Controlled Drug Release Dosage Forms"

_pharmaceutics, 2021, doi:10.3390/pharmaceutics13040543_

Round 1

Reviewer 1 Report

Overall, the work is well-written and of interest to the readers of pharmaceutics. However, there are few comments that should be addressed before publication. 

  1. Please, mention the supplier, maker, city and country form all the reagents and equipments and software used.
  2. It is key that authors quantify total drug loading of ibuprofen inside the printed scaffolds. Otherwise it is difficult to understand the potential of this technique to load 3D printed structures.  
  3. How do you know which is your 100% in your dissolution studies?
  4. The discussion is poor. Please comment on the challenges of this technique, the efficiency of the process and also compare the drug loading you can achieve with those obtained by other authors for example employing passive diffusion (see paper: Personalised 3D printed medicines: Optimising material properties for successful passive diffusion loading of filaments for fused deposition modelling of solid dosage forms), hot melt extrusion (3D printed spherical mini-tablets: Geometry versus composition effects in controlling dissolution from personalised solid dosage forms)...
  5. Please, add the application of this technique of drug loading. Which administration route are you envisaging?

Reviewer 2 Report

The manuscript (pharmaceutics-11398229) entitled "Supercritical fluid technology for the development of 3D printed controlled drug release dosage forms"  is well written with a reasonable set of experiments and sound discussion. Indeed, the manuscript should be revised as per the following suggestions for further improvement.

  1. It is suggested to include the physicochemical characteristics of the ibuprofen which favor the design and development of such novel drug delivery system exploiting 3D printing technology.
  2. If possible, include a standard/control (marketed product of ibuprofen) to compare the dissolution profile of the optimized developed formulation system of ibuprofen. 
  3. Authors are advised to include future directions of this platform technology in drug delivery in the conclusion section of the manuscript for the researcher working in this area. 

Reviewer 3 Report

Authors proposed a paper entitled “Supercritical fluid technology for the development of 3D 2 printed controlled drug release dosage forms” for the publication on the Pharmaceutics Journal, Mdpi.

This paper has a good scientific soundness since it combines the technology of 3D printing with the production of drug carriers.

The manuscript is well written and deserves to be published.

I have some issues listed here.

Line 10. define API in the abstract

Authors may add an abbreviation list according to the guidelines of this Journal

Abstract in my opinion has little numeric information. I would add them in order to achieve readers interest.

Line 13. “The 3D printed drug carriers were then loaded with Ibuprofen”. say what was the percentage in mass of drug/carrier matrix.

Line 19 “ forms with individually tailored controlled drug release”. please give some numerical information about drug release; for example, the maximum release time reached.

Line 36. there is a huge literature about the production of liposomes using supercritical fluids, for example DELOS, DESAM, SuperLip, and sometimes SAS process. Authors maybe could cite some of these.

Line 38. “ were developed[7],” there is a missing space before [7].

Line 45. Is the reference “Türk et al.” related to [10]? However, [10] is at the end of this paragraph.

Line 47. Same problem with “Wischum erski et al.”. Is it referred still to [10]? please check this.

Line 55. the literature about 3D printing technologies could be enlarged. for example, the laser sintering is a pioneeristic one for drug delivery.

Line 70. “with temperature-70 sensitive drugs.” please add a reference here.

Line 76. “we developed scaffold like structures”. I suggest not using personal forms in the manuscript.

Table 2 could be unified in 1 group of 2 columns.

Line 127. “was flooded” I would say “filled”

Line 134. “Because of the strong temperature drop, the drug loaded carriers need to remain in B1 for 45 min after the pressure decrease, to 135 prevent condensation of water on the carriers.”. However, with a strong depressurization, also the temperature decreases a lot, causing Joule Thomson effect. could you comment this deeply?

Figure 4. I suggest adding a reference bar in this figure, in order to have confirmation about macro dimensions of these printed drug carriers.

Line 192. “Figure 4.).” there is a point more after “4”.

Description of the SEM is missing in the section Materials and Methods.

Same observation of confocal microscopy. the description of the instrument/technology is required in the proper section.

Line 209. Please correct “direct vicintiy”

Figure 10. Some error bars exceed the 100% release. Is it correct?

I would comment Figure 10 also in terms of percentage of drug released. the final “point” of the first 3 carriers is almost overlapping, whereas the final point of carrier 4 is much lower. could you comment deeply this phenomenon, according to release time? is this due to smallest pores?

Thank you

Round 2

Reviewer 3 Report

Authors provided a revised version of this paper. 

They responded to my issues and now the paper deserves to be published.

Please, check the format of Table 3: the ending line is missing.

Check the format of the added paragraph in Discussion section.

Check also the "Conclusion" headline position.

Thank you